# A Systematic Review of Clinical Trials Assessing Sexuality in Hysterectomized Patients

**DOI:** 10.3390/ijerph18083994

**Published:** 2021-04-10

**Authors:** Laura Martínez-Cayuelas, Pau Sarrió-Sanz, Antonio Palazón-Bru, Lidia Verdú-Verdú, Ana López-López, Vicente Francisco Gil-Guillén, Jesús Romero-Maroto, Luis Gómez-Pérez

**Affiliations:** 1Urology Services, University Hospital of Vinalopó, 03293 Elche, Alicante, Spain; martinezcayuelaslaura@gmail.com; 2Urology Services, University Hospital of San Juan de Alicante, 03550 San Juan de Alicante, Alicante, Spain; pausarrio@gmail.com (P.S.-S.); llanais@hotmail.com (A.L.-L.); jromeroma@coma.es (J.R.-M.); luisgope@gmail.com (L.G.-P.); 3Department of Clinical Medicine, Miguel Hernández University, 03550 San Juan de Alicante, Alicante, Spain; vgil@umh.es; 4Urology Services, Hospital of Marina Baixa, 03570 Villajoyosa, Alicante, Spain; lidiaverdu@hotmail.com

**Keywords:** hysterectomy, sexuality, abdominal, vaginal

## Abstract

In hysterectomized patients, even though there is still controversy, evidence indicates that in the short term, the vaginal approach shows benefits over the laparoscopic approach, as it is less invasive, faster and less costly. However, the quality of sexual life has not been systematically reviewed in terms of the approach adopted. Through a systematic review, we analyzed (CRD42020158465 in PROSPERO) the impact of hysterectomy on sexual quality and whether there are differences according to the surgical procedure (abdominal or vaginal) for noncancer patients. MEDLINE (through PubMed), Embase, Cochrane Central Register of Controlled Trials, ClinicalTrials.gov and Scopus were reviewed to find randomized clinical trials assessing sexuality in noncancer patients undergoing total hysterectomy, comparing vaginal and abdominal (laparoscopic and/or open) surgery. Three studies that assessed the issue under study were finally included. Two of these had a low risk of bias (Cochrane risk of bias tool); one was unclear. There was significant variability in how sexuality was measured, with no differences between the two approaches considered in the review. In conclusion, no evidence was found to support one procedure (abdominal or vaginal) over another for non-oncological hysterectomized patients regarding benefits in terms of sexuality.

## 1. Introduction

Hysterectomy, whether via the open abdominal, laparoscopic, or vaginal procedure, is the most common gynecological surgery in developed countries after cesarean section. The choice of one procedure over another depends on a range of factors [1,2,3,4,5,6].

Historically, the most frequently used method has been open abdominal surgery, particularly in cases of tumors or enlarged uterus. Vaginal access is a very practical option as long as uterine mobility and size allow it. Laparoscopic surgery is increasingly used, though inhibited by the learning curve associated with its technical difficulty [5].

Hysterectomy has been associated with a high rate of short- and long-term complications [7,8,9,10,11]. In the long term, sexual dysfunction (persistent or recurrent reduction of sexual desire, arousal, orgasm, along with the presence of pain) may occur, which, although not life-threatening, can significantly limit the quality of life of the patient [7,12,13].

The etiology of sexual dysfunction is very complex, as it involves psychological, genetic, and anatomical factors [14]. The uterus is part of a women’s sexual identity, and any pathology that affects it can entail psychosexual problems [7,13,15], such as decreased libido and changes in genital sensitivity [15,16]. Hysterectomy can cause changes in vascularization and innervation, along with decreased secretion and vaginal shortening, possibly giving rise to dyspareunia [2,4,7]. The unsatisfactory quality of sexual activity prior to the intervention also seems to have a negative influence on postoperative results [3]. Contrastingly, some studies report instances of hysterectomy performed in cases of benign pathologies, such as large prolapses or dysmenorrhea, where sexual function improves [2,4,7,13,14].

The procedure adopted is the most influential factor in general postoperative comorbidity. Although there is still debate in this area, a recently published systematic review comparing the choice of procedure in terms of short-term comorbidity indicated that the vaginal procedure was more beneficial, being less invasive, faster, and less costly. Sexual quality outcomes were not analyzed in this review [6].

Post-hysterectomy sexual dysfunction is a controversial topic that is eliciting increased clinical interest [9,17]. It would be interesting to know if the choice of the procedure has any effects on sexuality. Consequently, a systematic review was undertaken to evaluate the impact on the sexual quality of hysterectomy and whether there are differences relating to the surgical procedure [9,17].

## 2. Materials and Methods

### 2.1. Protocol

The protocol of this review was registered in the PROSPERO database (CRD42020158465). The guidelines of the statement on preferred reporting items for systematic reviews and meta-analyses (PRISMA) and the Cochrane Library were followed in its preparation [18,19]. The indications published in a measurement tool to assess systematic reviews (AMSTAR-2), which correspond to a systematic review with a low risk of systematic errors, have also been taken into account in the preparation of this review [20].

### 2.2. Selection Criteria

All the randomized clinical trials (RCTs) that compared total vaginal versus abdominal hysterectomy (laparoscopic and/or open) carried out on women requiring this type of intervention in terms of female sexuality were selected. The exclusion criteria include observational or non-randomized studies, those that did not take sexuality into account, and those that included cases of tumor pathology. Only RCTs were selected, as this type of study presents fewer systematic errors in assessing interventions [21]. No minimum or maximum time threshold for follow-up was established, as we were interested in short-, medium-, and long-term results.

### 2.3. Information Sources and Search

A bibliographic search was carried out in MEDLINE (through PubMed) and Embase databases, covering all studies published from their creation until December 2020. Articles in both Spanish and English were included, as these are the two languages in which the authors are proficient and for which abstracts for each article were included. The RCT filter was also incorporated into the study design. The following keywords were used: hysterectomy, abdominal, vaginal, sexuality, sexual behavior and sexual. The complete search equations are included in Appendix A. Additionally, two registries of RCTs were analyzed through hand-searching: Cochrane Library and ClinicalTrials.gov (accessed on 9 March 2021). A manual review was also carried out of the bibliography of all selected articles, and the Scopus portal (Elsevier: Amsterdam, North Holland, The Netherlands) was used to review scientific articles in which they were cited. Whenever doubts arose relating to individual studies, the authors were contacted, especially if the RCT was registered, labeled as completed, and there were no published results. It should also be noted that the lead author of this review is a specialist in the subject area. Finally, the gray literature was searched manually through Google, searching for unpublished reports in scientific journals that included RCTs and met the selection criteria of our review.

### 2.4. Study Selection

Titles and abstracts were reviewed independently by two researchers (LM and PS) to exclude those that did not meet the inclusion criteria. A third reviewer (LG) was available for instances of discrepancy but was not called upon to intervene. Once the abstracts were selected, the same process was repeated for the full text of the articles selected in the previous step.

### 2.5. Data Extraction

For each of the articles eventually selected, the two reviewers mentioned above extracted the following information using the same procedure: author, year of publication, population, study design, intervention performed, control group, sample size, outcomes and form of measurement, results and effect of the intervention (beneficial, neutral, harmful). Data and clinical variables that could be confounders for sexuality were extracted: age (years, mean), menopause (%), hormone therapy (%) and adnexectomy (%).

### 2.6. Risk of Bias

The Cochrane Library tool was used to measure the risk of bias in the papers selected [19]. This evaluates seven domains of the RCTs: random sequence generation (selection bias), allocation concealment (selection bias), blinding of participants and researchers (performances bias), blinding of outcome assessment (detection bias), incomplete outcome data (attrition bias), selective reporting (reporting bias) and other biases. The assessment for each item can be classified as low risk of bias, uncertain (unknown or insufficient information to assess bias) or high risk of bias. Each domain was evaluated according to the Cochrane guidelines. To reduce subjectivity in certain questions, this evaluation was carried out both in pairs and independently by the same researchers, who participated in the other review stages. Finally, the global risk of bias was evaluated, following the “worst score counts” principle, according to which each article was evaluated according to the lowest rating obtained in the seven previous domains.

### 2.7. Statistical Analysis

A descriptive analysis of the data obtained from each publication was carried out, though the small number of studies identified meant that a meta-analysis was not possible.

## 3. Results

Figure 1 is a flowchart of the stages of the systematic review. The search found 106 studies, 49 and 57, respectively, from MEDLINE and Embase. A further three were added from the Cochrane Library trials register [1,22,23]. After eliminating duplicates, 90 studies were evaluated by title and abstract, and of these, 12 subsequently underwent full-text analysis [1,16,22,23,24,25,26,27,28,29,30,31]. In this analysis, nine articles were eliminated for the following reasons: vaginal procedure not included: (*n* = 6) [24,25,26,27,28,30], sexuality not analyzed (*n* = 2) [22,29], or because they were systematic reviews (*n* = 1) [23]. Hence, three papers were eventually included in our systematic review [1,16,31].

Table 1 shows the main characteristics of the articles selected [1,16,31]. These compare, through an interventional study, the sexuality impact after different hysterectomy procedures (vaginal versus open abdominal or laparoscopic) in patients with a history of tumor pathology being excluded in all of them [1,16,31]. The mean age of the patients was around 45 years. No information about hormone therapy was available in every paper. Regarding menopause and adnexectomy, the evaluated papers showed great variability, and in some cases, the authors did not report the prevalence of those conditions. The design of all the studies was RCT, with one being multicenter. The total sample size ranged from 41 in the smallest series to 504 in the largest. The latter study included another RCT, in addition to the one in our review, comparing abdominal and laparoscopic hysterectomy [31]. In all cases, the outcome was different and assessed by questionnaires.

The assessment of sexuality in the selected works was heterogeneous, as different questionnaires and scales were used. Garry et al. [31] used “The sexual activity questionnaire” for sexuality [34], as well as the 12-item short-form health survey (SF-12) for quality of life [32], and the body image scale for body image [33]. In summary, the SF-12 score is a multipurpose short-form generic measure with 8 components: health status, including physical functioning, role—physical, bodily pain, general health, energy/fatigue, social functioning, role—emotional and mental health. Each component is assessed using a Likert scale, and the total score ranges from 0 to 100 points, where the higher score, the better overall mental and physical well-being [32]. The body image scale asks the patients about how they feel about their appearance and any changes that may have resulted from their disease or treatment. It is a 10-item scale that assesses several dimensions of body image (affective, behavioral, and cognitive). Like the SF-12, this scale uses Likert items to be added in a total score, which goes from 0 to 30 points, where 0 points represent no affection in your body image and 30 the maximum affection [33]. The sexual activity questionnaire is a multipurpose questionnaire for sexual function. It is based on 3 domains: hormonal status (six items assessing the hormonal status and whether or not women are sexually active), reasons for sexual inactivity (seven possible reasons for sexual inactivity are listed) and sexual functioning (desire, frequency, satisfaction, dryness of vagina and penetration pain). The questionnaire is based on binary and Likert items to obtain a score [34]. The working group led by Wierrani used a questionnaire created by the authors themselves [16], in which sexual desire and genital sensitivity were assessed preoperatively and at three, six, and 12 months after surgery, using a Likert-type scale, with −2 being the worst possible value and +2 the best. Finally, Candiani et al. [1] posed a dichotomous variable in terms of sexuality, asking sexually active patients if they continued to maintain sexual activity at months one, six, and 12 following surgery. To sum up, no article identified differences between the procedure used and sexuality [1,16,31], indicating a neutral effect.

Table 2 summarizes the risk of bias according to the Cochrane classification [19]. In the article by Wierrani et al. [16], the randomization sequence, as well as the allocation and study protocol, are not sufficiently clear; therefore, in our opinion, in these three fields, the risk of bias is unclear, and the same would be true for the global protocol, as we choose the worst result [19]. The remaining two articles were classified globally as low-risk [1,31], as each of their individual sections received this rating.

## 4. Discussion

### 4.1. Summary

The three articles eventually selected for review are of good quality overall, with two qualifying as having a low risk of bias [1,31] and one as unclear [16]. The analysis concludes that the impact on the sexuality of one procedure over another is neutral.

### 4.2. Limitations

It is possible that selection bias occurred, for instance, in the exclusion of articles that met the inclusion criteria and had none for exclusion. However, as the peer review was carried out independently across all phases, this bias is reduced as far as possible for the studies reviewed. On the other hand, the failure to include all available databases could also be a reason for such a bias, so an extension of the search to other databases, adapting the search equations used, could be helpful (Appendix A). It might be thought that the systematic review would have reduced validity by virtue of being based on only three heterogeneous studies [1,16,31], but it should be noted that the RCTs are of sufficient quality, and the conclusions of each are clinically relevant. Taking into account that the question of sexuality has not been previously analyzed systematically [6], we consider that this study provides important information for routine clinical practice in patients of this type and could be the basis for future studies. Finally, there was missing information in clinical variables, which could have influenced sexuality. However, we must consider that we are assessing clinical trials, which have randomized the participants, to surmise the groups were comparable.

### 4.3. Comparison with Existing Literature

According to the literature reviewed, there is controversy around sexuality after hysterectomy. Most patients experience improvement after both types of procedures [24,25]. Between 15 and 30% experience deterioration, which could be a secondary effect related to both the anatomical alteration and psychological factors relating to the loss of the female organ, among others [9,35].

As regards prolapse, some studies show improvements in sexual quality after resolution of this condition [29]. Hysterectomy can play an important role in treating prolapse since it can be used for its correction either as a single technique or in association with repair via autologous tissues or synthetic meshes. The question arises as to whether performing a hysterectomy in treating prolapse correction affects sexual quality. In general, the results are again neutral, finding no differences relating to the performance of hysterectomy over repair techniques [36]. Some studies have identified better results in terms of sexuality when the uterus is preserved, associating this with better results for desire, arousal, and orgasm, related to the fact that the uterus is part of female sexuality both at an anatomical and an emotional [37]. As regards the use of hysterectomy as prophylactic surgery in response to possible tumor development over time, this procedure should not be systematically recommended [38], as the functional and anatomical benefits are not clear and, in general, women prefer uterine preservation whenever good functional results are guaranteed [39].

The systematic review does not appear to establish differences in terms of sexuality for the different hysterectomy procedures, taking the vaginal procedure as a reference. Consequently, although the results in terms of sexuality do not seem conclusive when comparing the two techniques, there may be other clinical factors that are more clear-cut, as described in the systematic review by Lee et al., which advocates the vaginal route as a less invasive procedure, that is faster and less costly [6].

If we only examine the abdominal procedure, comparing open and laparoscopic surgery, the results in terms of sexuality are less consistent. Garry et al. [31] find (in the section of the study devoted to this procedure) greater sexual activity at six weeks following laparoscopic surgery. Similarly, Ayoubi et al. [9] show sexual improvement levels rising by up to 80% after hysterectomy, with better results after laparoscopic surgery. However, other studies do not identify differences between the two techniques [25,26].

Vaginal shortening after hysterectomy was assessed as a possible cause of dyspareunia. Polat et al. [24] demonstrate less vaginal shortening with the laparoscopic procedure than open abdominal surgery. However, this does not translate into differences in sexuality between the two groups.

Broadly speaking, hysterectomy plays a crucial role in the management of gynecological pathology as an important tool in treating benign and malignant pathologies. In those that are benign, there are unknowns still to be resolved. The first is whether sexuality improves or deteriorates after surgery. The second is whether hysterectomy entails identifying the procedure that provides the greatest benefits, both overall and specifically in terms of sexuality. Finally, there was significant interest in the area of sexuality in recent years, placing women themselves and their preferences front and center of all surgical decisions.

### 4.4. Implications for Research and Clinical Practice

Sexuality is important to women’s quality of life, and patients increasingly seek the resolution of their pathologies without adverse effects in this area. The clinician is duty-bound to adjust treatment according to the needs and preferences of each patient. This review highlights the importance of anticipating functional results following hysterectomy and of considering the procedure on a patient-by-patient basis so as to obtain optimal clinical and quality of life results. That is why our results are applicable when the surgical planning of patients takes place. Given that this systematic review does not demonstrate differences between the procedures, the decision should be based on other factors, such as the experience of the surgical team, patient preference, possible complications, recovery time, expense, menopause, hormone therapy, adnexectomy, history of psychological issues and indication for surgical therapy like chronic pelvic pain syndrome, among others. Consequently, this decision should be individualized, bearing in mind the number of factors, which could affect sexuality vs. surgery. For this reason, more observational longitudinal studies should be carried out to assess this relevant clinical question.

We have found scant scientific literature on this topic, as most studies do not consider the vaginal route [24,25,26,27,28,30]. On the other hand, the heterogeneity of the design and the limited number of studies found suggest that further research is warranted that would incorporate sufficient sample size, validated measurement instruments, homogeneous protocols, and the inclusion of the vaginal procedure. This would allow an assessment of recovery levels following the different procedures using a meta-analysis of clinical trials, which was not possible in our study for the reasons described above. Additionally, all factors that may relate to differences between the two procedures should be included, such as complications, health costs, length of hospital stay, among others, to set up subgroups of patients that share similar postoperative characteristics and expectations so as to be able to offer personalized treatment options.

## 5. Conclusions

The review indicates that there are no differences in sexuality based on surgical procedures. However, these results should be interpreted with caution due to the low number of articles analyzed and their design differences. For this reason, and in the light of the relevance of the research premise, further studies with adequate sample sizes, similar designs, and standardized evaluation tools are called for.

## Figures and Tables

**Figure 1 ijerph-18-03994-f001:**
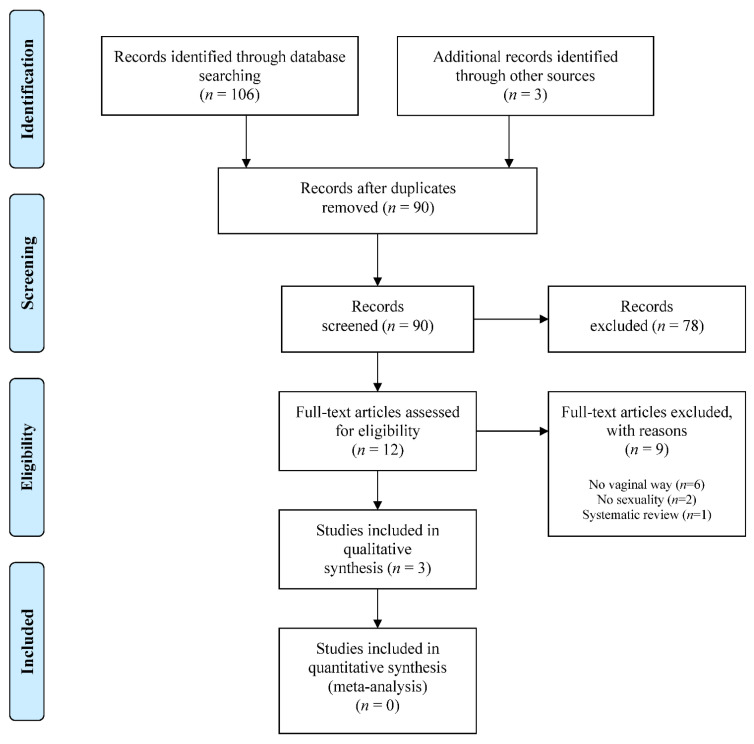
Preferred reporting items for systematic reviews and meta-analyses (PRISMA) flowchart of the systematic review.

**Table 1 ijerph-18-03994-t001:** Main characteristics of the articles included in the review.

References	Population	Age (Years, Mean)	Menopause (%)	Hormone Therapy (%)	Adnexectomy (%)	Design	Intervention	Control	N Intervention	N Control	Outcome	Measurements	Effect
Garry et al. 2004 [31]	Non-malignant hysterectomy	40.8 (VH) 40.9 (LH)	Unknown	Unknown	Unknown	RCT multicentric	VH	LH	168	336	QoL (physical and mental) Body Image Sexual activity	SF-12 score [32] Body image scale [33] The sexual activity questionnaire [34]	Neutral
Wierrani et al. 1995 [16]	Non-malignant hysterectomy	43.3	0	Unknown	0	RCT unicentric	VH	AH, LAVH, CASH	14	27	Libido and genital sexual sensitivity	Authors’ questionnaire (Likert-type scale)	Neutral
Candiani et al. 2009 [1]	Non-malignant hysterectomy	51.2 (VH) 48.9 (LH)	30	Unknown	59 (VH) 56.7 (LH)	RCT unicentric	VH	LH	30	30	Sexual problems	Dichotomous response	Neutral

RCT (randomized clinical trial). VH (vaginal hysterectomy). LH (laparoscopic hysterectomy). LAVH (laparoscopic-assisted vaginal hysterectomy). CASH (celioscopy-assisted hysterectomy). QoL (quality of life). SF-12 Score (short-form 12 items score).

**Table 2 ijerph-18-03994-t002:** Risk of bias according to the Cochrane classification. A final global risk column was added, following “the worst score counts” principle.

	Random Sequence	Allocation	Blinding of Participants	Blinding of Outcome	Incomplete Data Outcome	Selective Reporting	Other Bias	Global
Garry et al. [31] 2004	Low	Low	Low	Low	Low	Low	Low	Low
Wierrani et al. [16] 1995	Unclear	Unclear	Low	Low	Low	Unclear	Low	Unclear
Candiani et al. [1] 2009	Low	Low	Low	Low	Low	Low	Low	Low

## Data Availability

Data is contained within the article.

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
