# Peer review of "A Systematic Review of Clinical Trials Assessing Sexuality in Hysterectomized Patients"

_ijerph, 2021, doi:10.3390/ijerph18083994_

Round 1

Reviewer 1 Report

Martinez-Cayuelas et al. reported a systematic review of clinical trials assessing sexuality in hysterectomized patients. The paper is overall well written. Although physicians and gynecologists should be women's advocate, woman sexuality has been commonly undermined and has been commonly in the bottom of the complain and management list. I want to congratulate the authors for looking into this subject since there is really not much data on this topic. As stated by the authors, this is one of the limitations of this study since it is a systematic review of prior studies.

I would like the suggest the following:

1-Introduction is well written and informative to the reader, however I suggest that authors include background information on the clinical or psychosocial sexuality assessment tools that are used in the studies they included in their results section.

2-I am not aware of a well studied screening tool for sexuality assessment. This should be expressed in a better/clearer way. The results section can be written better especially the second and third paragraph to help the reader follow easier.

3-I suggest that patient demographics and variables that can be confounders for sexuality such as age, if in menopause or not, on hormone therapy or not, type of the surgery like if bilateral salphigoophorectomy is included during the hysterectomy or not etc should be addressed. If this information can be obtained from the three studies reviewed that would be very helpful since surgical techniques are being compared. However if this retrospective data not obtainable this should be included in the discussion section. It should be emphasized that there are other variables besides the hysterectomy approach are important in sexuality.

4-I agree the authors comment on section 4.4 that this subject definitely should be included in the counseling session prior to the surgery. The decision should be individualized. Sexuality can be affected by variety of other factors such as history of psychological issues, the reason for surgical therapy like chronic pelvic pain syndrome, and all other variables that I partially included in #3. I understand that there is no perfect analysis and assessment especially for a such challenging and difficult topic however I think these issues should be discussed and addressed in the discussion.

Author Response

Martinez-Cayuelas et al. reported a systematic review of clinical trials assessing sexuality in hysterectomized patients. The paper is overall well written. Although physicians and gynecologists should be women's advocate, woman sexuality has been commonly undermined and has been commonly in the bottom of the complain and management list. I want to congratulate the authors for looking into this subject since there is really not much data on this topic. As stated by the authors, this is one of the limitations of this study since it is a systematic review of prior studies.

We appreciate the reviewer´s comments. Based on their suggestions, the following points were addressed:

I would like the suggest the following:

1-Introduction is well written and informative to the reader; however, I suggest that authors include background information on the clinical or psychosocial sexuality assessment tools that are used in the studies they included in their results section.

We have included background information  related to different scales that have been used in  the studies included in our systematic review (SF-12, Body Scale and the sexual activity questionnaire). This information has been included in Results section and now is written:  In summary, SF-12 score is a multipurpose short-form generic measure with 8 components: health status, including physical functioning, role-physical, bodily pain, general health, energy/fatigue, social functioning, role-emotional and mental health. Each component is assessed using a Likert scale and the total score ranges from 0 to 100 points, where the higher score the better overall mental and physical well-being.[32] The Body Image Scale asks the patients about how they feel with their appearance and any changes that may have resulted from their disease or treatment. It is a 10-item scale that asses several dimensions on body image (affective, behavioural and cognitive). As the SF-12, this scale uses Likert items to be added in a total score, which goes from 0 to 30 points, where 0 points represents no affection in your body image and 30 the maximum affection.[33] The sexual activity questionnaire is a multipurpose questionnaire for sexual functioning. It is based on 3 sections: hormonal status (six items assessing hormonal status and whether or not women are sexually active), reasons for sexual inactivity (seven possible reasons for sexual inactivity are listed) and sexual functioning (desire, frequency, satisfaction, dryness of vagina and penetration pain). The questionnaire is based on binary and Likert items to obtain a score”.

2-I am not aware of a well studied screening tool for sexuality assessment. This should be expressed in a better/clearer way. The results section can be written better especially the second and third paragraph to help the reader follow easier.

We agree with the reviewer and more information about the instruments to assess sexuality in the final papers of our systematic review (paragraph #3) has been included. On the other hand, paragraph #2 has been modified. New results as been added and paragraph has been rewritten.

Table 1 shows the main characteristics of the articles selected.[1,16,31] These compare, through an interventional study, the sexuality differences after surgery in different hysterectomy procedures (vaginal versus open abdominal or laparoscopic) with patients with a history of tumour pathology being excluded in all of them.[1,16,31] The mean age of the patients was around 45 years with no information about hormone therapy in the three papers. Regarding menopause and adnexectomy, the three papers showed great variability and in some cases the authors did not report the prevalence of some of those conditions.”

3-I suggest that patient demographics and variables that can be confounders for sexuality such as age, if in menopause or not, on hormone therapy or not, type of the surgery like if bilateral salphigoophorectomy is included during the hysterectomy or not etc should be addressed. If this information can be obtained from the three studies reviewed that would be very helpful since surgical techniques are being compared. However, if this retrospective data not obtainable this should be included in the discussion section. It should be emphasized that there are other variables besides the hysterectomy approach are important in sexuality.

We agree the reviewer and we have updated Methods section to address the new variables suggested (section 2.5), Table 1 and second paragraph in Results section. Furthermore, we have written a comment in Discussion section related to the missing data in the information extracted.

4-I agree the authors comment on section 4.4 that this subject definitely should be included in the counseling session prior to the surgery. The decision should be individualized. Sexuality can be affected by variety of other factors such as history of psychological issues, the reason for surgical therapy like chronic pelvic pain syndrome, and all other variables that I partially included in #3. I understand that there is no perfect analysis and assessment especially for a such challenging and difficult topic however I think these issues should be discussed and addressed in the discussion.

We have included the reviewer comments in section 4.4.

Reviewer 2 Report

Congratulations on the great manuscript. Unfortunately, I don't think the results are completely true to reality. Two issues cause me concern:
1. Why weren't pre-print results or gray literature like Scholar Google included? This limits the findings a lot and is probably one of the reasons why only 3 manuscripts were recovered;
2. The data is out of date. I suggest that the search be redone on all bases and updated until December 2020.

Author Response

Congratulations on the great manuscript. Unfortunately, I don't think the results are completely true to reality. Two issues cause me concern:

We appreciate the positive reviewer´s comments. Based on the reviewer suggestions the following points were addressed.

  1. Why weren't pre-print results or gray literature like Scholar Google included? This limits the findings a lot and is probably one of the reasons why only 3 manuscripts were recovered;

The grey literature was searched (lines #94-96): “Finally, the grey literature was searched manually through Google, searching for un-published reports in scientific journals that included RCTs and met the selection criteria of our review.”  An update of the grey literature showed no new reports.

  1. The data is out of date. I suggest that the search be redone on all bases and updated until December 2020.

We agree with the reviewer and the search has been updated until December 2020. We have accordingly modified Methods (section 2.3), Results (first paragraph) and Figure 1.

Round 2

Reviewer 1 Report

I would like to thank Authors regarding the revised manuscript. The methodology and results sections are easier to follow. The background information on the questionnaire and scales is very helpful. 

Thank you

Reviewer 2 Report

The authors responded to all suggestions.